# A scoping review of medication self-management intervention tools to support persons with traumatic spinal cord injury

Lauren Cadel[1,2], Stephanie R. Cimino[3,4], Glyneva Bradley-Ridout[5], Sander L. Hitzig[3,4,6], Tanya L. Packer[7,8], Lisa M. McCarthy[1,2,9,10,11], Tejal Patel[9,12], Aisha K. Lofters[10,11], Shoshana Hahn-Goldberg[1,13], Chester H. Ho[14], Sara J. T. Guilcher[1,2,3,4,15]*

1 Leslie Dan Faculty of Pharmacy, University of Toronto, Toronto, ON, Canada, 2 Institute for Better Health, Trillium Health Partners, Mississauga, ON, Canada, 3 Rehabilitation Sciences Institute, Temerty Faculty of Medicine, University of Toronto, Toronto, ON, Canada, 4 St. John's Rehab Research Program, Sunnybrook Research Institute, Sunnybrook Health Sciences Centre, Toronto, ON, Canada, 5 Gerstein Science Information Centre, University of Toronto, Toronto, ON, Canada, 6 Department of Occupational Science and Occupational Therapy, Temerty Faculty of Medicine, University of Toronto, Toronto, ON, Canada, 7 Schools of Occupational Therapy and Health Administration, Dalhousie University, Halifax, NS, Canada, 8 Department of Rehabilitation, Radboud University Medical Centre, Nijmegen, The Netherlands, 9 Schlegel-University of Waterloo Research Institute of Aging, Waterloo, ON, Canada, 10 Department of Family and Community Medicine, University of Toronto, Toronto, ON, Canada, 11 Women's College Research Institute, Toronto, ON, Canada, 12 University of Waterloo School of Pharmacy, Kitchener, ON, Canada, 13 OpenLab, University Health Network, Toronto, ON, Canada, 14 Division of Physical Medicine & Rehabilitation, Department of Clinical Neurosciences, Foothills Medical Centre, Calgary, AB, Canada, 15 Institute of Health Policy, Management and Evaluation, University of Toronto, Toronto, ON, Canada

* sara.guilcher@utoronto.ca

**Data Availability Statement:** All relevant data are within the manuscript and its Supporting Information files.

## Abstract

### Background

Persons with traumatic spinal cord injury (SCI) use multiple medications (polypharmacy) to manage the high number of secondary complications and concurrent conditions. Despite the prevalence of polypharmacy and challenges associated with managing medications, there are few tools to support medication self-management for persons with SCI.

### Objective

The purpose of this scoping review was to identify and summarize what is reported in the literature on medication self-management interventions for adults with traumatic SCI.

### Methods

Electronic databases and grey literature were searched for articles that included an adult population with a traumatic SCI and an intervention targeting medication management. The intervention was required to incorporate a component of self-management. Articles were double screened and data were extracted and synthesized using descriptive approaches.

**Funding:** The research was supported by the Craig H. Neilsen Foundation Psychosocial Research Studies and Demonstration Projects Grant (#855615, received by SJTG). The funders had no role in study design, data collection and analysis, decision to publish, or preparation of the manuscript.

**Competing interests:** The authors have declared that no competing interests exist.

## Results

Three studies were included in this review, all of which were quantitative. A mobile app and two education-based interventions to address self-management of SCI, medication management, and pain management, respectively, were included. Only one of the interventions was co-developed with patients, caregivers, and clinicians. There was minimal overlap in the outcomes measured across the studies, but learning outcomes (e.g., perceived knowledge and confidence), behavioural outcomes (e.g., management strategies, data entry), and clinical outcomes (e.g., number of medications, pain scores, functional outcomes) were evaluated. Results of the interventions varied, but some positive outcomes were noted.

## Conclusions

There is an opportunity to better support medication self-management for persons with SCI by co-designing an intervention with end-users that comprehensively addresses self-management. This will aid in understanding why interventions work, for whom, in what setting, and under what circumstances.

## Introduction

Traumatic spinal cord injury (SCI) is a life altering condition. Persons with SCI commonly experience secondary complications and concurrent conditions (e.g., pain, spasticity, neurogenic bowel and bladder complications, respiratory complications, depression) [1, 2]. Medications are commonly used for the prevention, management, and treatment of these secondary complications and chronic conditions experienced [3–8]. For example, laxatives (constipation), analgesic-narcotics (acute or chronic pain), anticonvulsants (neuropathic pain), skeletal muscle relaxants (spasticity, pain), serotonergics (depression), tricyclic antidepressants (depression, neuropathic pain), sedatives (anxiety, sleep disorders), and anxiolytics (anxiety) are frequently prescribed medication classes post-injury [6].

Unsurprisingly, the number of medications taken on a daily basis can dramatically change as a result of a SCI, with one study reporting a three-fold increase in the number of medications taken by individuals after sustaining a SCI [9]. This results in a high prevalence of persons experiencing polypharmacy (i.e., taking multiple medications, typically more than 5 per day) [10]. Rates of polypharmacy post-injury range from 31% to 87% (variation in rates attributable to differences in how polypharmacy was defined and types of medications included) [5, 6, 10–12]. Importantly, polypharmacy has been associated with an increased incidence of adverse drug events among persons with SCI [13]. Polypharmacy has also been linked to increased impaired cognition and higher rates of hospitalization and mortality among the general population [14].

Persons with SCI have described numerous challenges managing multiple medications including difficulties integrating medications into their everyday lives (e.g., remembering to take medications at specific times; obtaining refills in a timely manner), often feeling overwhelmed when dealing with medication regimens, side effects, and communicating with healthcare providers [15–17]. Recent qualitative research with persons with SCI identified a psychological resistance to taking medications, particularly for individuals who identified as being healthy and had not taken medications prior to their injury [17]. Pain medications are of particular concern for this population, with individuals in this study describing anxiety

regarding safety of both short and long-term use, in addition to fears about their pain medications losing effectiveness over time. Further concerns were raised about side effects of medications (e.g., fatigue, constipation, addiction) and medication regimen complexity (taking numerous medications on different schedules, integrating into daily life). Importantly, this study [17], along with other qualitative research, [15] identified a lack of self-management supports around medications among persons with SCI.

Medication self-management extends beyond medication adherence [18] and can be conceptualized as a range of tasks, skills, and behaviours associated with an individual's capability, opportunity, and motivation to navigate the physical, social, and cognitive lifestyle factors, changes, and consequences inherent in taking, or choosing not to take, medications in everyday life. These tasks include having the knowledge and related confidence to deal with medical, emotional, and role management, as well as the core skills of problem-solving, decision-making, seeking formal and informal supports, self-tailoring, goal-setting, optimizing social interactions, and engaging in activities, as they relate to managing medications [19, 20]. In terms of medication self-management, as described above, there has been very limited research conducted to date on potential benefits of medication self-management programs for individuals with physical impairments who experience polypharmacy.

Despite the prevalence of polypharmacy and challenges associated with managing medications, there remains a lack of tools to support persons with SCI with medication self-management. Therefore, the purpose of this scoping review was to identify and summarize what is reported in the literature on medication self-management interventions for adults with traumatic SCI. More specifically, we aimed to identify how the interventions were designed and delivered, the components of the interventions, and the measures used to evaluate the implementation or outcomes of the interventions.

## Methods

This scoping review was conducted according to the methodology outlined by Peters and colleagues [21]. The Preferred Reporting Items for Systematic Reviews and Meta-Analyses extension for scoping review (PRISMA-ScR) was used as the reporting guideline (see S1 Table) [22].

### Protocol and registration

The protocol for this scoping review was registered on OSF Registries (https://osf.io/89qha).

### Eligibility criteria

This scoping review was part of a larger research study that aims to develop and evaluate a toolkit for medication self-management for persons with SCI. For the larger scope of work and to ensure relevant literature was identified to inform the design and development of a toolkit, we also included other neurological populations. Specifically, we included studies about person who have experienced stroke as part of our eligible population. Herein, we present the results of the articles specific to SCI, with the stroke-specific findings to be published separately (no articles included both stroke and SCI). We have opted to separate the results of the two populations to allow for a full exploration of the findings, gaps, and implications and translation of knowledge for both SCI and stroke, as fundamental differences were noted.

Therefore, to be included in this scoping review, articles were required to: (1) include an adult population (aged 18+) with traumatic SCI; (2) include an intervention aimed at modifying or improving medication management; and (3) incorporate a component of self-management into the intervention. To meet the first inclusion criteria, at least 50% of the participants had to be over the age of 18 with SCI. To meet the second criteria, we defined medication

management as the tasks, skills, and behaviours associated with an individual's capability, opportunity, and motivation to navigate the physical, social, and cognitive lifestyle factors, changes, and consequences inherent in taking, or choosing not to take, medications. To meet the third criteria, the intervention had to include a component of self-management based on how we operationalized it for this scoping review–having the knowledge and related confidence to deal with medical, emotional, and role management, or the core skills of problem-solving, decision-making, seeking formal and informal supports, self-tailoring, goal-setting, optimizing social interactions, and engaging in activities, as they relate to managing medications.

Articles were excluded if they met at least one of the following criteria: (4) opinion pieces and narrative reviews; (5) conference abstracts; (6) study protocols; or (7) if we were unable to access the full-text. We excluded conference abstracts and study protocols to ensure all included articles presented finalized results. This allowed us to fully examine intervention implementation characteristics and associated outcomes.

## Search methods

The searches were developed by an academic health sciences librarian (GBR). Five electronic databases were searched on March 11<sup>th</sup>, 2022: MEDLINE (Ovid Interface), EMBASE (Ovid Interface), CINAHL Plus (EBSCOhost Interface), APA PsycINFO (Ovid Interface), and Clarivate Web of Science. The Ovid MEDLINE search was PRESS peer-reviewed by a second academic librarian prior to search translations [23]. The searches were constructed using the concepts of (traumatic SCI OR stroke) AND self-management AND medication. The search was translated into the databases using each platforms' command language and controlled vocabulary, where applicable. No limits were applied on the searches. The full database search strategies, copied and pasted exactly as run, can be found in S2 Table. The electronic database searches were supplemented by grey literature searches conducted June 17, 2022 on relevant websites, including the World Health Organization, Spinal Cord Injury Research Evidence (SCIRE), and Praxis Spinal Cord Institute.

## Selection process

Articles from the electronic database searches were deduplicated using EndNote X8 [24]. Following deduplication, Covidence was used to facilitate the screening processes. Three reviewers (LC, SRC, SJTG) screened 150 titles and abstracts to ensure good agreement (>80% agreement) [25]. The reviewers had over 95% agreement. No revisions or clarifications to the eligibility criteria were required, and the remaining titles and abstracts were double screened. All disagreements were resolved through consensus. Following the title and abstract screen, two reviewers (LC, SRC) completed a test screen of ten full-text articles to ensure good agreement and that all criteria were being interpreted and applied in the same way. The reviewers had 90% agreement and the remaining full-text articles were double screened. All disagreements were resolved through consensus. The PRISMA flow diagram documenting the records identified, included, and excluded can be found in Fig 1.

## Data charting process

A data extraction table was created in Microsoft Excel to facilitate the extraction process. Two team members (LC, SRC) tested the extraction table and conducted a spot check of an article to confirm all information was extracted consistently and accurately. No revisions to the extraction table were made and the same two team members independently extracted

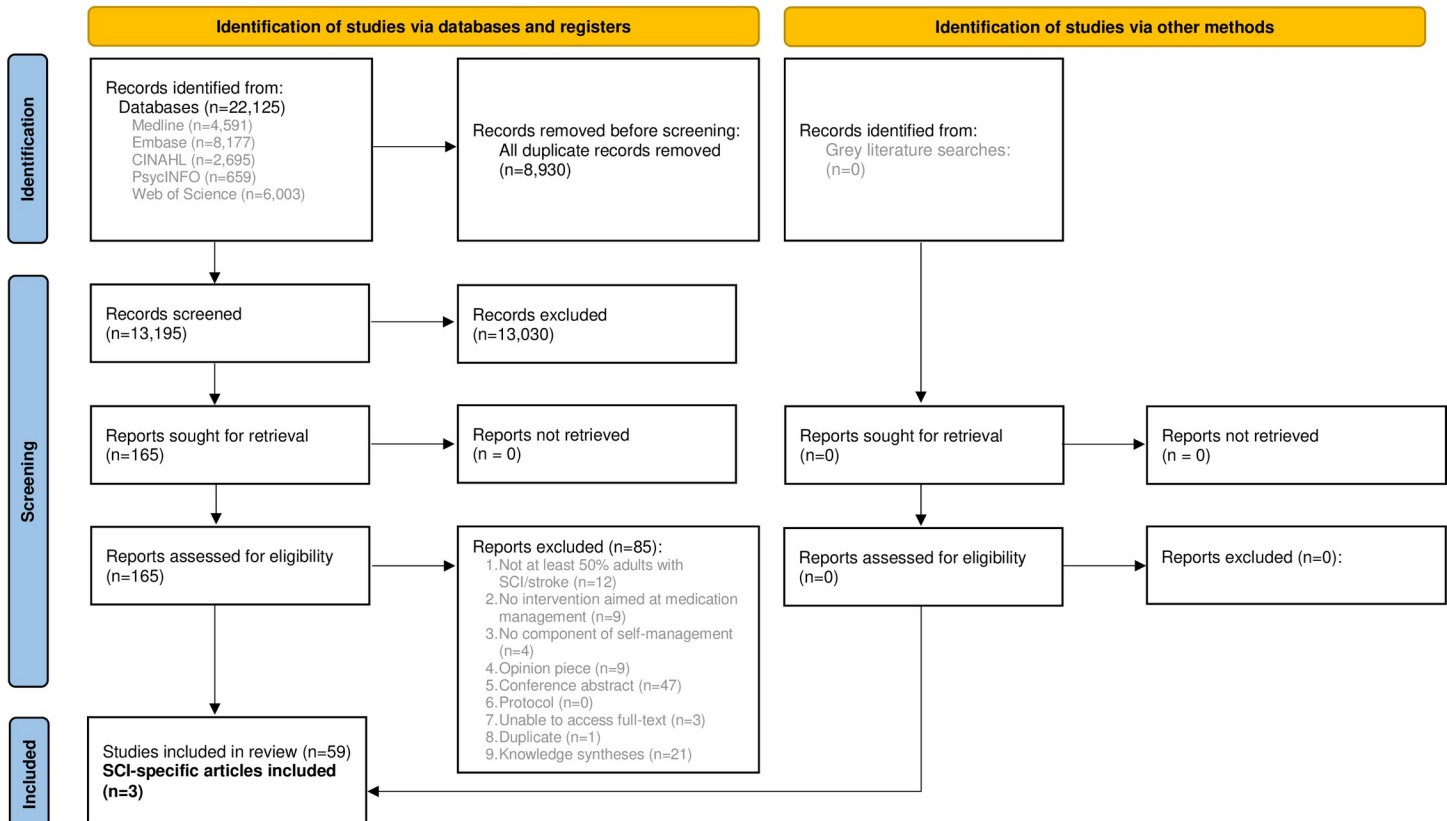

*From:* Page MJ, McKenzie JE, Bossuyt PM, Boutron I, Hoffmann TC, Mulrow CD, et al. The PRISMA 2020 statement: an updated guideline for reporting systematic reviews. BMJ 2021;372:n71. doi: 10.1136/bmj.n71. For more information, visit: http://www.prisma-statement.org/

**Fig 1. PRISMA flow diagram of included articles.**

information from the remaining full-text articles. All data were single-extracted and the final, completed extraction table was reviewed by both team members who completed the extraction.

## Data items

The extracted variables included: general article information (title, authors, year of publication, journal, funding), study characteristics (objective, type of population, method of data collection, study design, theoretical orientation, eligibility criteria, outcomes, country, setting), intervention characteristics (description, content, frequency, duration, single or multi-component, format, tailoring, modifications, method of delivery, setting), population characteristics (sample size, age, sex, gender, ethnicity/race, income, education, marital status, household composition, employment status, comorbidities), study outcomes and findings (results and key findings, conclusions). The Template for Intervention Description and Replication (TIDieR) checklist guided which data items to extract related to the intervention characteristics [26].

## Synthesis methods

For this scoping review, we only present the results of the articles for SCI as the results of the stroke articles will be published separately. Data from the included SCI articles were

synthesized descriptively, including the study designs, countries, years of publication, population characteristics, intervention characteristics, and intervention outcomes. The intervention outcomes were categorized by one team member (LC) as a learning outcome, behavioural outcome, or clinical outcome. Learning outcomes were defined as the knowledge, skills, abilities, attitudes, and understanding achieved through participation in the intervention [27]. Behavioural outcomes were defined as actions that individuals consciously engaged or did not engage in [28]. Clinical outcomes were defined as changes in health, function, or quality of life [29]. The TIDieR checklist was used to help structure the results [26]. A critical appraisal of articles was not conducted and is not a requirement of scoping reviews [22].

## Results

### Study selection

A total of 22,125 articles were identified from the database searches, with 13,195 remaining following deduplication. At the title and abstract level of screening, 13,030 articles were excluded, so 165 remained for full-text screening. Of the 165 full-text articles screened, 85 were excluded. We searched the reference lists of knowledge syntheses (systematic and scoping reviews; n = 21) that met the inclusion criteria, but did not include them in data extraction or synthesis, leaving 59 articles. For the purposes of this scoping review, we separated the stroke and SCI articles, resulting in 3 included articles.

### Study characteristics

Characteristics of included articles are displayed in Table 1. All of the studies were quantitative [30–32]; two used a prospective design [30, 32], and one was a retrospective analysis of a quasi-experimental design [31]. Two of the three studies had an SCI comparator group [31, 32]; one compared the intervention group to a control group [31] and the other to a non-response group [32]. All of the included articles were published within the last five years and each was conducted in a different country–Canada [30], the United States [31], and Korea [32].

### Population characteristics

The overall sample sizes of the studies varied (n = 20 [30], n = 207 [31], n = 109 [32]). All of the articles reported the sex of the participants, rather than gender, and all included a higher proportion of male participants than female [30–32]. Two of the articles reported the participants' education level, with the majority of participants having at least a high school education [30, 31]. One of the articles reported the participants' ethnicity and level of income [31] and another reported the participants' employment status and living environment (e.g., city, suburban, rural) [30].

### Intervention characteristics

Characteristics of interventions, aligning with the TIDieR checklist, are displayed in Table 2. The three included articles described and implemented three different interventions–SCI Storylines [30], an Educational Medication Management Program [31], and a Pain Management Program for SCI [32]. SCI Storylines was a mobile app designed to improve self-management related to SCI [30], while the latter two were education-based interventions designed to improve knowledge specific to medication management [31] and pain medication use [32]. Only one of the interventions (Educational Medication Management Program [31]) specifically targeted medication management. None of the articles described specific theory informing the interventions; however, the Educational Medication Management Program was

**Table 1. Characteristics of included articles (n = 3).**

| Author (year), Country | Objective | Study Design | Participant Characteristics | Sample Size | Intervention and Control Descriptions | Descriptive and Outcome Measures | Key Findings |
|---|---|---|---|---|---|---|---|
| MacGillivray et al. (2020), Canada [30] | • To examine the feasibility of the implementation and evaluation of a self-management app (SCI Storylines) for spinal cord injury (during inpatient rehabilitation and post-discharge) | Prospective longitudinal study | • Persons (age range 22–81) with an American Spinal Injury Association Impairment Scale (AIS) grade of A, B, C, or D • Injury type: 75% traumatic • Sex: 85% male • Ethnicity: not reported | 20 | Intervention: A newly developed app for self-management post-SCI (incorporated tools, tracking mechanisms, and journaling features) Control: N/A | • Feasibility indicators: recruitment rate, retention, intervention adherence, usability, adverse events, use/ administration of outcome measures • Spinal Cord Independence Measure-III self-report version • New General Self Efficacy Scale • Technology Readiness Index 2.0 • Custom Likert scale for importance of key areas of self-management relating to SCI • Hospital Anxiety and Depression Scale | • Inpatient rehabilitation was a feasible time to introduce the self-management app • Challenges with adherence and retention after discharge were experienced |
| Oyesanya et al. (2020), United States [31] | • To develop and test the efficacy of an intervention for medication management (Educational Medication Management Program), delivered prior to discharge from inpatient rehabilitation • To improve perceived confidence and knowledge for post-discharge medication management | Retrospective analysis of a practice re-design using quasi-experimental study | • Persons diagnosed with SCI, receiving inpatient rehabilitation and planned to be discharged home • Injury type: not reported • Sex: 58% male • Ethnicity: 63% White, 27% Black, 9% more than one | 207 | Intervention: 45-minute educational intervention consisting of a medication management video, walk-through of website, pillbox and medication scheduling exercise Control: Received usual care (discharge instructions and medication information), only completed post-discharge data collection (by telephone) | • Adapted existing medication management and adherence instruments to develop study-specific instrument (perceived knowledge and confidence for medication management, medication management issues) | • Significant improvement in perceived knowledge and confidence • Patients and family members can benefit from medication management intervention delivered pre-discharge (increased perceived knowledge and confidence) |
| Shin et al. (2017), Korea [32] | • To explore the impact of education (disease pathophysiology, drug mechanisms, side effects) on pain medication use through a Pain Management Program for SCI | Prospective intervention study | • Persons 20+ years of age, with SCI and neuropathic pain, taking pain medication • Injury type: 70% traumatic • Sex: 69% male • Ethnicity: not reported | 109 | Intervention: 6-week treatment program delivering education on natural course of the injury, physiology of SCI-related pain, pain medication information Control: Non-response group (no changes or an increase in the number of medications over study program) | • Visual Analog Scale for pain intensity • International Spinal Cord Injury Pain Classification • ASIA motor sub-scores • Functional Independence Measure • Spinal Cord Independence Measurement III mobility section • Beck Depression Inventory | • Pain management education can reduce medication side effects and pain, while improving patient stability |

**Table 2. TIDieR checklist.**

| TIDieR Checklist Item | SCI Storylines [30] | Educational Medication Management Program [31] | Pain Management Program for SCI [32] |
|---|---|---|---|
| Why | • A mobile app designed to facilitate self-management for persons with SCI | • An educational intervention designed to improve perceived knowledge and confidence for medication management post-discharge | • An educational pain management program designed increase knowledge and reduce pain medication use |
| What | • Consisted of 18 tools to help with the self-management of SCI including tracking, and journaling features<br>• Training sessions were delivered (how to navigate the tablet and the app) | • Consisted of detailed information, a video, walkthrough of the website, pillbox, and medication schedule exercises<br>• Included written and visual information, detail on medication labels, and strategies for medication reminders | • Consisted of weekly informational and feedback sessions (nature of the injury, pain physiology, types, dose, use, side effects, cost of pain medications)<br>• Patients received simple, descriptive handouts on their medication indications, mechanisms, and side effects |
| Who Provided | • Trainer (researcher team) | • Rehabilitation nurse educators | • Physiatrist, rehabilitation specialist |
| How | • In-person and technology (mobile app) | • Technology (videos and website) | • In-person |
| Where | • Rehabilitation hospital and community | • Inpatient rehabilitation hospital and community | • Rehabilitation department in hospital |
| When and How Much | • Training sessions (dependent on needs of participant), follow-ups from trainer once per week during inpatient rehabilitation and 1–2 times per month after discharge to community for 3 months | • One time educational session | • Once a week for 6 weeks |
| Tailoring | • Yes–Customized training sessions based on participant needs | • Not reported | • Not reported |
| Modifications | • Not reported | • Not reported | • Not reported |
| How Well | • *Behavioural Outcomes*: More data entry in the app occurred during rehabilitation than in the community<br>• *Learning Outcomes*: Improved bowel self-management confidence, trends toward improvements in bladder, autonomic dysreflexia, and pain management confidence (admission to discharge)<br>• *Clinical Outcomes*: N/A | • *Behavioural Outcomes*: The intervention group participants were more likely to use a pillbox and medication list to manage their medications<br>• *Learning Outcomes*: Improved perceived knowledge and perceived confidence for medication management (baseline to post-test, not maintained)<br>• *Clinical Outcomes*: N/A | • *Behavioural Outcomes*: N/A<br>• *Learning Outcomes*: N/A<br>• *Clinical Outcomes*: Significant reduction in number of individuals taking two or more types of pain medications, pain score decreased significantly from baseline, improved motor, sensory, and functional scores at follow-up |

developed using evidence-based literature on medication adherence [31]. This intervention was also developed in consultation with a panel of interdisciplinary healthcare providers who had experience with rehabilitative care. Similarly, SCI Storylines was developed using an iterative process that involved patients, caregivers, and clinicians [30]. This publication did not describe how the Pain Management Program was developed [32]. All articles described who was responsible for delivering the intervention–trainers (research team) [30], rehabilitation nurse educators [31], and a physiatrist and rehabilitation specialist [32].

The Pain Management Program was delivered in-person [32], the Educational Medication Management Intervention was technology-based [31], and SCI Storylines included in-person and technological components [30]. The in-person intervention (Pain Management Program) consisted of weekly information sessions about the course of the injury, SCI-related pain, and pain medication [32]. There was also an opportunity for participants to ask questions and provide feedback. The technology-based intervention (Educational Medication Management Intervention) included a medication management video, walk-through of the website, and pillbox and medication scheduling exercises [31]. Detailed written and visual information was provided to participants about medications (safety, off-label use, storage, disposal), medication-taking (adherence, pillboxes, schedules, medication list), communication with doctors, online resources, and reminder strategies. The intervention delivered with both in-person and

technological components (SCI Storylines) consisted of an app and face-to-face training sessions [30]. The app contained topics related to the self-management of SCI, including: bowel, bladder, skin, autonomic dysreflexia, orthostatic hypotension, urinary tract infections, spasticity, physical activity, goals, confidence, pain, fluid intake, fatigue, equipment, and neurological status. Incorporating tools, trackers, and journaling features, the app allowed participants to record information related to the listed topics, as well as their medication use, symptoms, and daily vital signs. All interventions were initiated in hospital or inpatient rehabilitation and ranged in duration from 6 weeks to 3 months. One of the articles tailored the informational sessions to meet the needs of the participants [30] and none of the articles described modifying the intervention during the course of the study.

## Intervention outcomes

There was minimal overlap in the outcomes measured in the included articles. The studies assessed learning outcomes, behavioural outcomes, and clinical outcomes. Across the three studies, the results of the interventions varied. In terms of learning outcomes, the Educational Medication Management Intervention improved participants' perceived knowledge and perceived confidence (assessed by a study-specific outcome measure) for medication management from baseline to post-test; however, the improvements were not maintained at 60-day follow-up [31]. SCI Storylines also impacted learning outcomes, with improved bowel self-management confidence and trends towards improved bladder, autonomic dysreflexia, and pain management confidence from admission to discharge [30]. Behaviour change with data entry in the app was seen as individuals entered data more times on average while in rehabilitation compared to in the community [30]. Participants in the Educational Medication Management Intervention treatment group participants were more likely than control group participants to use a pillbox and a medication list to manage their medications [31]. In terms of clinical outcomes, education provided in the Pain Management Program for SCI significantly reduced the number of individuals who took two or more types of pain medications [32]. Pain scores significantly decreased compared to baseline and improvements were noted in motor, sensory, and functional scores at follow-up compared to baseline [32].

## Discussion

The purpose of the scoping review was to identify interventions for medication self-management for persons with SCI. Overall, we found a lack of literature on this topic. Based on review of the three included articles, we identified the need for an intervention that (1) specifically targets and supports medication self-management for adults with SCI; (2) is co-designed in consultation with persons with SCI; and (3) comprehensively assesses quantitative and qualitative outcomes of the intervention.

This scoping review identified three different interventions, one mobile app to improve self-management related to SCI [30] and two education-based interventions to improve knowledge specific to medication management [31] and pain medication use [32]. The mobile app focused on the self-management of SCI more broadly and incorporated a medication tool and tracker, but medication self-management was not the primary focus [30]. The education-based medication management intervention focused on increasing perceived knowledge and perceived confidence for post-discharge medication management, while also incorporating more detailed information on obtaining, understanding, and taking medication [31]. Lastly, the education-based pain management intervention centred on increasing knowledge and reducing pain medication use [32]. None of the identified interventions addressed medication self-management comprehensively (e.g., integrating a range of tasks, skills, and behaviours

related to taking, or choosing not to take, medications). There was limited assessment of the impact of medications on the day-to-day lives of persons with SCI, which has been previously highlighted as a challenge [15–17, 33]. It is important for interventions to use a holistic approach to incorporate all areas of medication self-management. This includes the range of tasks, skills, and behaviours associated with an individual's ability to navigate the multi-faceted nature of medication-taking. Related to managing medications, interventions should address knowledge and confidence with problem-solving, decision-making, seeking formal and informal supports, self-tailoring, goal-setting, optimizing social interactions, and engaging in activities [19, 20, 34]. Each of the included interventions addressed certain areas of self-management, but none were comprehensive. Munce and colleagues have provided key considerations for implementing self-management programs for persons with SCI, which include identifying and addressing individual, programmatic, and environmental level factors [35]. There is potential to draw on this work and apply it to future medication self-management interventions in this population.

One of the included articles described the involvement of patients, caregivers, and clinicians in the development of the intervention [30], which can be used as an example for future intervention design. The SCI Storylines mobile app was designed using an iterative process that involved these stakeholders throughout the project stages. Importantly, this intervention demonstrated improved learning and behavioural outcomes, with significantly higher bowel self-management confidence, trends for improved bladder, autonomic dysreflexia, and pain management confidence from admission to discharge, and more data entry in the app during rehabilitation [30]. The meaningful involvement of end-users throughout the intervention plan, design, implementation, and adaptation is key to the co-design process [36]. While ensuring the perspectives of end-users are integrated, co-design offers a number of other benefits, including: relevant, applicable, and acceptable research questions and objectives, improved experiences and emotional outcomes (sense of accomplishment, confidence), increased knowledge, skills, and understanding related to the research process, and improved buy-in [36]. Despite these reported benefits, the co-design process also presents challenges that should be acknowledged, such as increased time and financial resources, potential tensions between researchers and end-users, feelings of tokenism, and trade-offs between research rigor and incorporating end-users' suggestions [36]. Strategies for mitigating these challenges and improving the usability, effectiveness, and acceptability of co-designed interventions have been developed [31]. Strategies include: involving diverse participants, ensuring co-design is the best approach, identifying champions to support implementation, enabling inclusive involvement, managing power relations, managing expectations, ensuring role clarity and ownership of the process, providing updates on implementation, capturing and sharing the impact, and celebrating successes [37]. These strategies can help facilitate a successful and positive co-design experience. Furthermore, with limited integration of the patient and caregiver voice into current medication self-management interventions for persons with SCI, this is a critical a gap that warrants future research.

The three included articles used quantitative study designs. We found that there was minimal overlap in the outcomes measured across the studies. Included studies assessed learning outcomes (e.g., perceived knowledge and confidence), behavioural outcomes (e.g., medication management strategies, data entry in app), and clinical outcomes (e.g., number of medications, pain scores, functional outcomes). Given the relatively small sample sizes and limited overlap of the measured outcomes, more research is needed to better understand the effectiveness of these interventions on learning, behavioural, and clinical outcomes.

Despite only including three studies, there were some positive results that may support the use of medication self-management interventions for those with SCI. Notable improvements

in perceived knowledge and confidence, based on a study-specific (unvalidated) instrument, and a reduction in the use of multiple pain medications were reported. Even with these positive outcomes, there is a need to more comprehensively assess both quantitative and qualitative outcomes of interventions for medication self-management for this population. According to the Medical Research Council's framework for developing and evaluating complex interventions, more emphasis should be placed on qualitative and mixed methods evaluations [38]. Qualitative research provides important contextual information about the interventions that cannot be understood through quantitative measures. For example, qualitative research can help explain how and why an intervention works, or does not work, as well as who it works for, when, and in what setting [38]. These insights are important when developing, implementing, and refining interventions because they highlight more than the effectiveness of the intervention, including what could be changed and/or adapted, what can be removed, and what needs to remain. Given the lack of qualitative and mixed methods studies on this topic, further research is needed to better understand what medication self-management interventions work, for whom, in what setting, and under what circumstances.

## Gaps and opportunities for future research

Overall, this scoping review highlighted the lack of research that exists on interventions related to medication self-management for persons with SCI and the numerous gaps to be addressed with future work. Aligning with the findings from this review, some suggested areas of future research include: (1) co-designing an intervention that comprehensively addresses medication self-management with persons with SCI, caregivers, and healthcare providers to ensure the perspectives of all stakeholders are both captured and integrated; and (2) further investigating both quantitative and qualitative outcomes of medication self-management interventions for persons with SCI. There is potential to build on these three studies in order improve learning, clinical, and behavioural outcomes and to better support medication management through a comprehensive intervention that incorporates the key components of self-management. A companion review with the results of the stroke articles will be published to fully discuss the findings and implications specific to stroke, as well as to ensure adequate translation of knowledge for each population.

## Limitations

This scoping review has a few limitations to note; it is possible that relevant articles were missed as we excluded opinion pieces, conference abstracts, study protocols, and articles in which we could not access the full-text through our library or interlibrary loan system. However, the University of Toronto has an extensive catalogue of resources, and is the largest academic library in Canada [39]. We limited our search strategy to traumatic SCI, so it is possible that interventions specific to non-traumatic SCI were not identified or included. Self-management is not a consistently defined term across disciplines, so it is possible that articles were missed due to the terms searched. We attempted to mitigate this by using comprehensive terms in our search for self-management and self-management related tasks, skills, and behaviours. Lastly, a critical appraisal of articles was not conducted; however it is not a requirement of scoping reviews [22].

## Conclusions

This scoping review identified a limited number of interventions targeting medication self-management for adults with SCI. The findings highlight an opportunity to further enhance support for medication self-management for persons with SCI by developing a more

comprehensive intervention that addresses all areas of self-management, co-designing interventions so they include end-user perspectives, and investigating quantitative and qualitative outcomes of the intervention to better understand why interventions work, for whom, in what setting, and under what circumstances.

## Supporting information

**S1 Table. Preferred Reporting Items for Systematic reviews and Meta-Analyses extension for Scoping Reviews (PRISMA-ScR) checklist.**
(DOCX)

**S2 Table. Full search strategies for all electronic databases.**
(DOCX)

## Acknowledgments

We would like to thank Julia Martyniuk, a librarian at the University of Toronto, who peer reviewed the Ovid MEDLINE search strategy.

## Author Contributions

**Conceptualization:** Lauren Cadel, Stephanie R. Cimino, Sara J. T. Guilcher.

**Data curation:** Sara J. T. Guilcher.

**Formal analysis:** Lauren Cadel, Stephanie R. Cimino, Sara J. T. Guilcher.

**Funding acquisition:** Lauren Cadel, Stephanie R. Cimino, Sander L. Hitzig, Tanya L. Packer, Lisa M. McCarthy, Tejal Patel, Aisha K. Lofters, Shoshana Hahn-Goldberg, Chester H. Ho, Sara J. T. Guilcher.

**Investigation:** Sara J. T. Guilcher.

**Methodology:** Lauren Cadel, Stephanie R. Cimino, Glyneva Bradley-Ridout, Sara J. T. Guilcher.

**Project administration:** Lauren Cadel, Sara J. T. Guilcher.

**Supervision:** Sara J. T. Guilcher.

**Validation:** Sara J. T. Guilcher.

**Visualization:** Sara J. T. Guilcher.

**Writing – original draft:** Lauren Cadel, Stephanie R. Cimino, Glyneva Bradley-Ridout, Sara J. T. Guilcher.

**Writing – review & editing:** Lauren Cadel, Stephanie R. Cimino, Glyneva Bradley-Ridout, Sander L. Hitzig, Tanya L. Packer, Lisa M. McCarthy, Tejal Patel, Aisha K. Lofters, Shoshana Hahn-Goldberg, Chester H. Ho, Sara J. T. Guilcher.

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
