## [Decision Letter · Decision Letter 0]

10 Jan 2023

PONE-D-22-28883A scoping review of medication self-management interventions for persons with traumatic spinal cord injuryPLOS ONE

Dear Dr. Guilcher,

Thank you for submitting your manuscript to PLOS ONE. After careful consideration, we feel that it has merit but does not fully meet PLOS ONE’s publication criteria as it currently stands. Therefore, we invite you to submit a revised version of the manuscript that addresses the points raised during the review process.

We look forward to receiving your revised manuscript.

Kind regards,

Saeed Ahmed, MD

Academic Editor

PLOS ONE

Journal Requirements:

Reviewers' comments:

Reviewer's Responses to Questions

**Comments to the Author**

1. Is the manuscript technically sound, and do the data support the conclusions?

Reviewer #1: Yes

Reviewer #2: Yes

Reviewer #3: Yes

2. Has the statistical analysis been performed appropriately and rigorously? 

Reviewer #1: Yes

Reviewer #2: Yes

Reviewer #3: N/A

3. Have the authors made all data underlying the findings in their manuscript fully available?

Reviewer #1: Yes

Reviewer #2: Yes

Reviewer #3: Yes

4. Is the manuscript presented in an intelligible fashion and written in standard English?

Reviewer #1: Yes

Reviewer #2: Yes

Reviewer #3: Yes

5. Review Comments to the Author

Reviewer #1: Thank you for shedding light on this topic. There is a dearth of literation integrating the patient and caregiver voices, with the exception of some collaborative care model data. This paper has a remarkable potential to bring this issue into the mainstream through PLOS, but I suggest a minor revision.

Line 220-222

Consecutive sentences “All of the articles reported the sex of the participants, and all included a higher proportion of male participants than female [32-34]. None reported the participants’ gender.” may be misconstrued to be internally inconsistent. Consider re-writing for text flow.

Overall, the findings of this scoping review paper would be valuable to the discipline of collaborative care, hospice, and Bio-ethics, advocating for enhanced patient autonomy.

Reviewer #2: The article is an interesting read. Assimilating the available articles and comparing the three literatures that offer tools for self-management of medication was done tactfully, carefully separating the outcomes- per behavioral outcomes, learning outcomes and clinical outcomes were thoughtful.

The only recommendation would be modifying the title: The scoping review of medication self-management, provided an initial thought to me- if patients with SCI struggled with self-medication challenges, only to realize after reading the article that they struggle managing their polypharmacy/medications independently without appropriate tools.

One recommendation for the title could be- Scoping Review of Self-Management Intervention tools utilized for treatment of persons with Traumatic Spinal Cord Injury.

Reviewer #3: The manuscript by Cadel et al. studies the important topic of self-management interventions in spinal cord injury (SCI) considering the chronicity of the condition, the short- and long-term impact on general health, and the quality of life of the patient and their caregivers.

Strengths of the paper:

- The abstract describes the findings of the study as well as suggestions on what can be done.

- The introduction to the article is well organized.

- Methods and results are clearly described.

- The discussion and conclusion further elaborate on the challenges in the treatment of patients with SCI and strategies for mitigating these challenges.

Some recommendations:

- As the authors have pointed out, self-management is not consistently defined. It may be worth adding a brief commentary on the findings of other publications to develop a model for self-management for SCI, for example, by Munce et al. (Munce, S.E.P., Webster, F., Fehlings, M.G. et al. Meaning of self-management from the perspective of individuals with traumatic spinal cord injury, their caregivers, and acute care and rehabilitation managers: an opportunity for improved care delivery. BMC Neurol 16, 11).

6. PLOS authors have the option to publish the peer review history of their article (what does this mean?). If published, this will include your full peer review and any attached files.

Reviewer #1: No

Reviewer #2: No

Reviewer #3: No

---

## [Author Response · Author response to Decision Letter 0]

7 Feb 2023

 - We have confirmed that the manuscript meets PLOS ONE’s style requirements and naming conventions. 

We have also reviewed the reference list to ensure it is complete and correct. No changes were made.

Reviewer #1: Thank you for shedding light on this topic. There is a dearth of literation integrating the patient and caregiver voices, with the exception of some collaborative care model data. This paper has a remarkable potential to bring this issue into the mainstream through PLOS, but I suggest a minor revision.

 - Thank you for this comment and for your recommendations.

Line 220-222

Consecutive sentences “All of the articles reported the sex of the participants, and all included a higher proportion of male participants than female [32-34]. None reported the participants’ gender.” may be misconstrued to be internally inconsistent. Consider re-writing for text flow. We have revised this section to improve flow. 

Overall, the findings of this scoping review paper would be valuable to the discipline of collaborative care, hospice, and Bio-ethics, advocating for enhanced patient autonomy.

 - Thank you for this comment.

Reviewer #2: The article is an interesting read. Assimilating the available articles and comparing the three literatures that offer tools for self-management of medication was done tactfully, carefully separating the outcomes- per behavioral outcomes, learning outcomes and clinical outcomes were thoughtful.

 - Thank you for this comment.

The only recommendation would be modifying the title: The scoping review of medication self-management, provided an initial thought to me- if patients with SCI struggled with self-medication challenges, only to realize after reading the article that they struggle managing their polypharmacy/medications independently without appropriate tools. One recommendation for the title could be- Scoping Review of Self-Management Intervention tools utilized for treatment of persons with Traumatic Spinal Cord Injury. 

 - Thank you for this suggestion – we have revised the title. 

Reviewer #3: The manuscript by Cadel et al. studies the important topic of self-management interventions in spinal cord injury (SCI) considering the chronicity of the condition, the short- and long-term impact on general health, and the quality of life of the patient and their caregivers.

 - Thank you

Strengths of the paper:

- The abstract describes the findings of the study as well as suggestions on what can be done. 

- The introduction to the article is well organized.

- Methods and results are clearly described.

- The discussion and conclusion further elaborate on the challenges in the treatment of patients with SCI and strategies for mitigating these challenges.

 - Thank you for these comments.

Some recommendations:

- As the authors have pointed out, self-management is not consistently defined. It may be worth adding a brief commentary on the findings of other publications to develop a model for self-management for SCI, for example, by Munce et al. (Munce, S.E.P., Webster, F., Fehlings, M.G. et al. Meaning of self-management from the perspective of individuals with traumatic spinal cord injury, their caregivers, and acute care and rehabilitation managers: an opportunity for improved care delivery. BMC Neurol 16, 11). 

 - Thank you for this recommendation, we have included some commentary from the work by Munce and colleagues into the discussion.

---

## [Decision Letter · Decision Letter 1]

27 Mar 2023

A scoping review of medication self-management intervention tools to support persons with traumatic spinal cord injury

PONE-D-22-28883R1

Dear Dr. Guilcher,   

We’re pleased to inform you that your manuscript has been judged scientifically suitable for publication and will be formally accepted for publication once it meets all outstanding technical requirements.

Kind regards,

Saeed Ahmed, MD

Academic Editor

PLOS ONE

Reviewers' comments:

Reviewer's Responses to Questions

**Comments to the Author**

1. If the authors have adequately addressed your comments raised in a previous round of review and you feel that this manuscript is now acceptable for publication, you may indicate that here to bypass the “Comments to the Author” section, enter your conflict of interest statement in the “Confidential to Editor” section, and submit your "Accept" recommendation.

Reviewer #1: All comments have been addressed

Reviewer #3: All comments have been addressed

2. Is the manuscript technically sound, and do the data support the conclusions?

Reviewer #1: Yes

Reviewer #3: Yes

3. Has the statistical analysis been performed appropriately and rigorously? 

Reviewer #1: Yes

Reviewer #3: (No Response)

4. Have the authors made all data underlying the findings in their manuscript fully available?

Reviewer #1: Yes

Reviewer #3: Yes

5. Is the manuscript presented in an intelligible fashion and written in standard English?

Reviewer #1: Yes

Reviewer #3: Yes

---

## [Editor Report · Acceptance letter]

11 Apr 2023

PONE-D-22-28883R1 

A scoping review of medication self-management intervention tools to support persons with traumatic spinal cord injury 

Dear Dr. Guilcher:

I'm pleased to inform you that your manuscript has been deemed suitable for publication in PLOS ONE. Congratulations! Your manuscript is now with our production department. 

Kind regards, 

on behalf of

Dr. Saeed Ahmed 

Academic Editor

PLOS ONE